# Genomic retargeting of p53 and CTCF is associated with transcriptional changes during oncogenic HRas-induced transformation

Michal Schwartz [1,4,5], Avital Sarusi Portugez[1,5], Bracha Zukerman Attia[1], Miriam Tannenbaum[1], Leslie Cohen[1], Olga Loza[1], Emily Chase[1], Yousef Turman[2], Tommy Kaplan [3], Zaidoun Salah[2] & Ofir Hakim [1✉]

Gene transcription is regulated by distant regulatory elements via combinatorial binding of transcription factors. It is increasingly recognized that alterations in chromatin state and transcription factor binding in these distant regulatory elements may have key roles in cancer development. Here we focused on the first stages of oncogene-induced carcinogenic transformation, and characterized the regulatory network underlying transcriptional changes associated with this process. Using Hi-C data, we observe spatial coupling between differentially expressed genes and their differentially accessible regulatory elements and reveal two candidate transcription factors, p53 and CTCF, as determinants of transcriptional alterations at the early stages of oncogenic HRas-induced transformation in human mammary epithelial cells. Strikingly, the malignant transcriptional reprograming is promoted by redistribution of chromatin binding of these factors without major variation in their expression level. Our results demonstrate that alterations in the regulatory landscape have a major role in driving oncogene-induced transcriptional reprogramming.

[1] The Mina and Everard Goodman Faculty of Life Sciences, Bar-Ilan University, Ramat-Gan, Israel. [2] Al-Quds-Bard College for Arts and Sciences, Al-Quds University, Abu Dis, Palestinian Terretories, Palestine. [3] School of Computer Science and Engineering, The Hebrew University of Jerusalem, Jerusalem, Israel. [4] Present address: Department of Molecular Genetics, Weizmann Institute of Science, Rehovot, Israel. [5] These authors contributed equally: Michal Schwartz, Avital Sarusi Portugez. ✉email: ofir.hakim@biu.ac.il

Altered gene expression programs are a major factor in the development and expansion of cancer cells. These aberrant transcription patterns support and promote biological processes that are required along tumorigenesis, such as proliferation, resistance to cell death, and induction of angiogenesis[1]. Transcriptional programs are driven by a complex network of transcription factors acting on regulatory DNA elements such as enhancers (reviewed in[2]). Although changes in transcriptional programs during tumorigenesis could be driven directly by DNA mutations, in many cases they are affected indirectly by various altered cellular pathways. Transcription factors may affect transcriptional programs due to changes in their DNA binding, without harboring mutations, or change in their expression levels. Much of the effort to characterize the pathways contributing to cancer development mostly focus on the identification of either upstream mutations[3] or downstream RNA and protein expression patterns of cancer cells[4]. These are indeed very powerful approaches to reveal important drivers of cancer development and characterize cancer types. However, demarcating the intermediate molecular events triggering gene expression programs holds great promise for generating novel and improved diagnostic tools and identifying relevant targets for therapeutic interventions[5,6]. Indeed, a recent pan-cancer analysis demonstrated global enhancer activation in most cancer types compared with matched normal tissues[7]. Thus studying the regulatory basis of transcription reprogramming associated with carcinogenesis can define a set of transcription factors that are involved in shaping the transcriptional landscape of the cancer cell.

Transcription programs are regulated by a complex network of transcription factors, cofactors, and chromatin regulators[2,8]. This regulation occurs via binding of transcription factors to regulatory elements, i.e., enhancers, in a sequence-specific manner, and recruitment of transcriptional coactivators or corepressors[2,8]. Enhancer sequences contain multiple binding sites for a variety of transcription factors, and mediate transcriptional control by their combinatorial binding[2,9]. This regulation is further modulated by the epigenetic status of enhancers. Therefore, mapping active enhancers can be key to delineating regulatory networks driving transcriptional reprogramming during the process of carcinogenesis. Active enhancer regions are associated with different molecular characteristics that allow their identification, such as histone modifications, specifically, H3K27ac and H3K4me1[10–13], and occupancy of transcriptional co-activators, such as Ep300[14]. Importantly, for an enhancer to be bound by transcription factors and other regulatory DNA binding proteins, it must be accessible, therefore, a technique that is commonly used to map active regulatory elements is the detection of open chromatin by sensitivity to nucleases[15,16]. ATAC-seq is a recently developed sensitive method to measure chromatin accessibility which enables mapping of active regulatory regions in a genome-wide manner[17].

A major challenge, however, for inferring mechanistic principles of transcriptional regulation, is the difficulty to assign specific regulatory elements to their distant target genes. Enhancers and their regulated target genes can be located hundreds of kbs apart[18]. Importantly, the spatial organization of genomic information is non-random and is a major regulatory component of gene transcription[19,20]. In recent years it was demonstrated that mammalian chromosomes are partitioned into units of internal high spatial connectivity[21]. These units, termed Topologically Associating Domains (TADs), are large (few tens of kb to 3 Mb) chromosomal units encompassing multiple genes and regulatory elements[22]. Although the borders between TADs are considered cell-type invariant, at the local scale, chromosomal contacts between genes and their regulatory elements (within TADs) show cell-type specificity[19]. Moreover, variation in gene expression within TADs is highly correlated during differentiation and response to external stimuli, suggesting that TADs insulate the activity of regulatory elements to specific genes[18,23,24]. Therefore, TADs offer a proper and applicable framework to couple between regulatory elements and their target genes and study dynamic transcriptional regulation.

Here we combined, transcriptome analysis, genome-wide mapping of active regulatory sites with chromosome topology profiles to delineate the regulatory network underlying transcriptional reprogramming during carcinogenic transformation induced by overexpression of oncogenic HRas in human mammary epithelial cells. Our results reveal major gene expression changes accompanied by significant alterations in the regulatory landscape upon this transformation process. Linking, in 3D, differentially expressed genes to the catalog of dynamic regulatory elements revealed two candidate transcription factors, p53 and CTCF that contribute to these transcriptional changes. Strikingly, these factors act by redistribution of their chromatin binding without major variation in their expression level. Moreover, this work highlights the power of the approach by which the transcriptional regulatory landscape can be analyzed in view of the 3D organization of the genome to reveal dynamic regulatory networks underlying cancer phenotypes.

## Results

**Mutations in the HRas oncogene are among the most frequent in human tumors.** To investigate transcriptional regulatory changes during early carcinogenesis, we introduced an oncogenic copy of HRas carrying a single point mutation at amino acid 12 (G12V) into MCF10A cells, a nontransformed, near-diploid, immortalized mammary epithelial cell line. These cells (from here on termed G12V) first undergo growth arrest but after a few weeks continue growing and exhibit higher proliferation rates and higher cell survival capacity compared to their MCF10A counterparts (Fig. 1a, b). The tumorigenicity of the G12V cells was tested by different means. Anchorage-independent survival and growth are hallmarks of carcinogenic cells. Indeed, G12V cells exhibit growth in soft agar (Fig. 1c) and adherence independent survival (Fig. 1d). Following HRas dependent transformation we noticed a morphological change towards a more mesenchymal morphology, we, therefore, tested invasive potential and found that the G12V cells harbor invasive properties as tested by Matrigel invasion assay (Fig. 1e). Finally, the examination of their tumorigenic potential in vivo shows that G12V cells are able, although with low penetrance, to form small tumors when injected into the mammary fat pad of Nod-SCID mice (Fig. 1f). Thus the introduction of G12V HRas oncogene into MCF10A rendered these cell tumorigenic.

In order to define the transcriptional changes induced by the introduction of G12V HRas oncogene in MCF10A cells, we performed RNA-sequencing comparing G12V and MCF10A cells. We identified 1144 up-regulated and 1136 down-regulated genes (>1.3 fold, $p < 0.05$) accompanying this transformation process (Fig. 2a and Supplementary Data 1). Down-regulated genes are enriched in functions of cell death ($p = 2.67E-18$) and apoptosis ($p = 2.53E-16$) (Fig. 2b). Up-regulated genes are enriched in functions of invasion ($p = 2.5E-50$), cell survival ($p = 1.42E-26$), and cell movement (1.64E-25) and include tumor-related genes (Fig. 2c). The enriched molecular pathways (Fig. 2d) also include some signaling pathways tightly related to breast cancer, such as IL-8 signaling[25] and p53 signaling[3,4]. The top enriched upstream factors that can explain these gene expression changes (Fig. 2e, upstream factors) also include p53. Thus the alteration of gene expression profiles accompanying the expression of the G12V HRas oncogene is indicative of tumorigenic process.

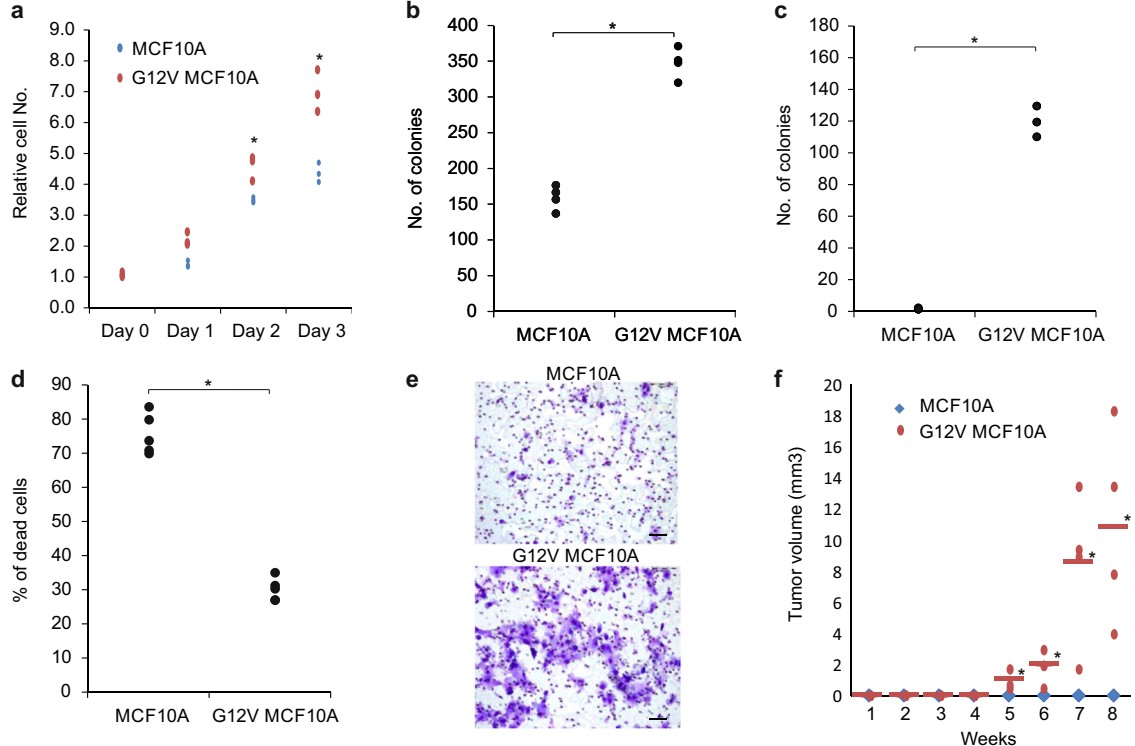

**Fig. 1 Characterization of G12V cells. a** Cell proliferation was measured by XTT assay. The relative number of cells, compared to day 0 is presented for MCF10A (blue line) and G12V cells (red line). **b** Cell survival was measured by colony formation assay. The number of colonies counted after 2 weeks of MCF10A and G12V cells are presented. **c** Anchorage-independent growth was measured by soft agar assay. The number of colonies formed for MCF10A and G12V cells is presented. **d** Resistance to anoikis was measured by anchorage-independent cell death assay. The percentage of cell death for MCF10A and G12V cells is presented. **e** Representative images of Boyden Chamber Matrigel invasion assay of MCF10A and G12V MCF10A cells (bar = 50 μm). **f** Tumor growth curve of mammary fat pad tumors in Nod-SCID mice injected with either MCF10A or G12V cells. Horizontal bars represent average. *indicates $p < 0.05$ T-test of at least three independent replicates.

In order to examine the changes in regulatory chromatin that dictate the transcriptional changes, we applied ATAC-seq which allows the discovery of accessible chromatin loci genome-wide. We identified 42,546 accessible loci in MCF10A cells and 46,367 in G12V cells. Using stringent analysis of differential peaks in MACS we identified a few thousands of increased and decreased peaks representing Differentially Accessible Regions (DARs, Fig. 3a, b). 5,355 were induced in G12V compared to MCF10A cells (gained DARs) while 7,589 were reduced (lost DARs). Overall, the distribution of accessible regions in the genome (Fig. 3c) is similar to what was previously reported[16] with ~30% of loci near promoters. Interestingly the proportion of DARs at promoters is much lower (5% in G12V and 8% in MCF10A) relative to the general distribution, indicating that the major changes in chromatin accessibility are at gene-distant regions (Fig. 3c). Examination of the connection between chromatin accessibility at promoters and gene expression reveals that up-regulated genes are more associated with gained accessibility while down-regulated genes associate more with loss of accessible regions (Fig. 3d). This reflects directional association between changes in gene expression and changes in their promoter accessibility.

The data points to TSS-distant accessible loci as potential key elements in regulating the transcriptional changes during oncogene-induced tumorigenesis in this model. Thus, in order to define regulatory pathways that are important in this process, it is essential to connect between DARs and their distant target genes.

The human genome is segmented into domains of the high frequency of internal long-range chromosomal associations, or topologically associating domains (TADs). TADs facilitate associations between their resident enhancers and gene promoters, while constraining inter-TAD contacts, thus constitute a spatial regulatory framework. Given that TADs are largely cell-type invariant, we tested the possibility to use available high-resolution Hi-C data from multiple cell types[26] for associating DARs with their gene targets. Using 4 C, we first tested for a few differentially expressing and a few stably expressing genes, whether their Hi-C defined topological domains from GM12878 cells are similar in G12V and MCF10A cells. As shown in Fig. 4a and Supplementary Fig. S1, indeed there is no change in the domains of the 5 genes we tested. We, therefore, took advantage of the comprehensive TAD database available[26] as a regulatory organizational framework, rather than genomic proximity, to link between TSS and regulatory elements (Fig. 4b)

The 3D organization of the genome has been shown to provide a functional framework for transcriptional regulation. Importantly, in line with this notion, we found that up-regulated and down-regulated genes are segregated between TADs and fall together in the same TAD much more rarely than expected by chance (Fig. 4c). Therefore, our data support that also in the case of oncogene-induced carcinogenic transformation, TADs are functionally relevant to transcriptional regulation, thus constitute an adequate framework to couple between transcriptional activity and regulatory loci as defined by chromatin accessibility. Importantly, cell-type-specific (MCF10A or G12V) DARs are enriched within TADs containing cell-type-specific expressed genes (Fig. 4d) suggesting that chromatin accessibility and genes are not only physically but also functionally linked within the 3D domains.

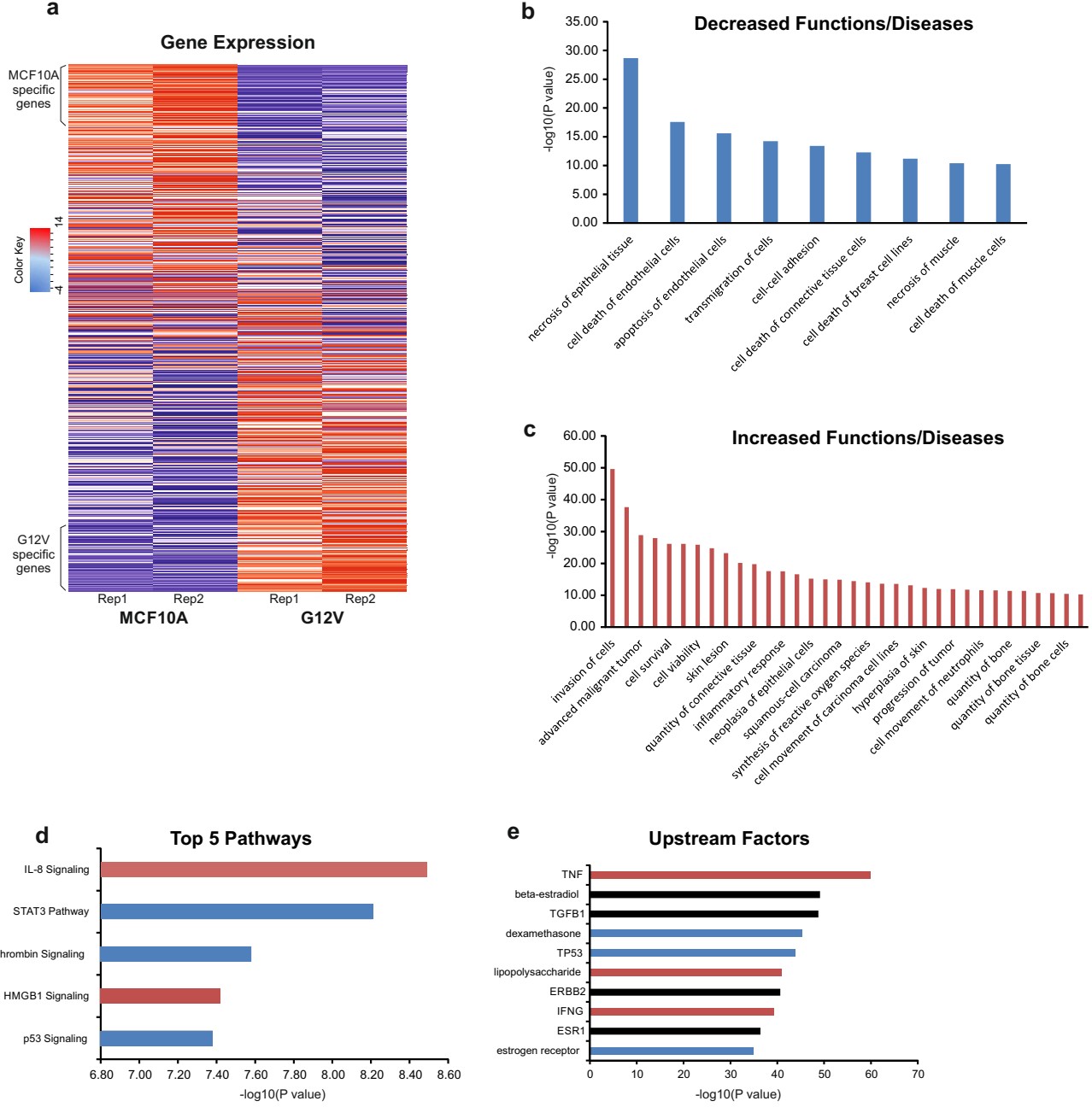

**Fig. 2 Transcriptional reprogramming induced by the G12V HRas oncogene. a** Heatmap showing gene expression of genes in MCF10A and G12V cells from two replicas. Rows were first ordered based on log2 fold change and then by expression value. **b, c** Functions and Diseases enriched in down-regulated genes (**b**) and up-regulated genes (**c**) with a log10 pVal >10. **d** Top five significantly affected pathways according to the DE genes. **e** Top 10 upstream factors which can explain the changes in gene expression. Red - Predicted activation; Blue- Predicted inhibition; Black- No specific direction of activity. Analysis in **b**–**e** was done using the Ingenuity Pathway Analysis software.

To discover candidate TFs involved in the transcriptional shift in this model of oncogene-induced transformation, we applied motif discovery analysis on different groups of regulatory sites defined by ATAC-seq, that are associated with differentially expressed genes in the same TAD.

Analysis of gained DARs, revealed enrichment for binding motifs of several TFs including ETV1 and Fli1 pro-oncogenes of the ETS family of transcription factors. Increased activity of ETS transcription factors was shown to be involved in all stages of tumorigenesis of several solid tumors, including prostate and breast cancer[27,28]. RNA-seq analysis uncovered several ETS factors that are differentially expressed in G12V cells and may

underlie some of the transcriptional alterations. One of the motifs that was found enriched in gained DARs is the CTCF motif. Interestingly, the CTCF motif is enriched in gained DARs located in TADs of both up-regulated and down-regulated genes in G12V cells (Fig. 5a) and in the global gained DARs dataset (8.97%). This suggests that the enrichment of the CTCF motif is associated with gained regulatory activity upon HRas oncogenic activation that is associated with both gene activation and gene repression. However, according to RNA-seq data, the levels of *CTCF* did not differ between G12V and parental MCF10A cells (Fig. 5B), nor CTCF protein levels, as tested by western blot analysis (Fig. 5b). Moreover, the integrity of *CTCF* coding region was not

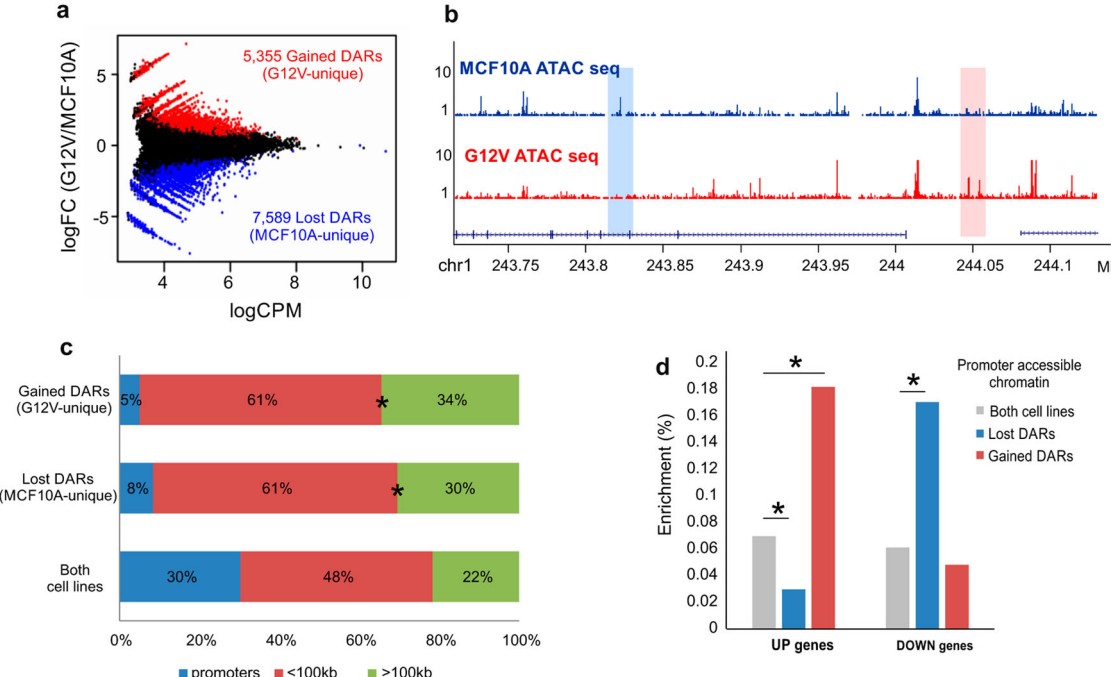

**Fig. 3 G12V HRas induced changes in the transcription regulatory landscape. a** MA plot displaying the mean normalized counts (Counts Per Million (CPM), *x*-axis) versus the log 2 fold change between MCF10A and G12V cells (*y*-axis), of the ATAC seq peaks. Red and blue points represent unique G12V (gained DARs) and unique MCF10A peaks (lost DARs), respectively, as declared by MACS analysis. **b** Representative examples of lost (blue box) or gained (pink box) DARs. Chromosomal coordinates in Mb of human hg19 genome build are indicated at the bottom. **c** Distribution of regulatory sites relatively to genes: promoter-proximal (+500 bp, −1000 bp from the gene's transcription start site (TSS), blue), mid-range (±100 kb, red) and far-range (green) in gained or lost DARs or accessible regions shared by both cell types. **p* < 0.001, proportional test. **d** % of regulatory sites near promoters of up-regulated genes in G12V cells (UP genes) or down-regulated genes in G12V cells (DOWN genes) from total regulatory sites near promoters. **p* < 0.001, proportional test.

compromised in G12V cells (Supplementary Fig. S4). To validate that indeed there is a change in CTCF binding upon activation of the G12V HRas oncogene, we performed chromatin immuno-precipitation coupled with next-generation sequencing (ChIP-seq) for CTCF in G12V cells and their MCF10A counterparts. This analysis revealed 30,642 ChIP-seq peaks in G12V cells and 22,376 peaks in MCF10A cells (Fig. 5c, Supplementary Fig. S5). As was predicted from the motif analysis, CTCF ChIP-seq data shows significantly higher CTCF binding in gained DARs compared to lost DARs located in TADs of both up-regulated and down-regulated genes (Fig. 5d, e). This strongly suggests the involvement of CTCF in regulating the differential gene expression upon HRas transformation in this system. Interestingly, this occurs via redistribution and probably increase of genomic CTCF binding upon HRas oncogene-induced transformation without a change in the level of *CTCF* expression.

In lost DARs, the most significantly enriched motif is the binding motif of p53 (Fig. 6a, GW 10.9%). This is in agreement with the fact that p53 is one of the top enriched upstream regulators with multiple of its targets being differentially expressed following HRas G12V overexpression in MCF10A cells (Fig. 2e). To confirm that variably expressed genes are regulated by p53 we asked whether genes that were down-regulated following G12V HRas transformation are responsive to p53 activation in MCF10A cells. To measure the likely direct transcriptional response to p53, RNA was extracted following a 4-hour Nutlin-3a treatment and sequenced. Gene expression changes were overall moderate. Importantly genes that were down-regulated following HRas induced transformation and were located within TADs harboring lost DARs that contain p53 motif showed the highest and significant response to p53 activation (Fig. 6d, and Supplementary

Fig. S3). We, therefore, checked the level of p53 for changes that could explain the mis-regulation of its target genes. Surprisingly *TP53* transcript and p53 protein levels, as well as its subcellular distribution, remain stable between G12V and MCF10A cells (Fig. 6b, c). Moreover, we confirmed that *TP53* gene is not mutated in the G12V HRas expressing cells (Supplementary Fig. S5). Thus the reduction in chromatin accessibility at putative p53 binding sites and mis-regulation of p53 target genes in G12V cells is not due to its downregulation or change in cellular localization. To assess directly whether indeed the landscape of p53 chromatin binding was altered in MCF10A and G12V cells, ChIP-seq was performed. Overall, 3260 binding sites of p53 were found in both cell types, with 465 and 565 sites specific for MCF10A and G12V cells, respectively (Fig. 6g, Supplementary Fig. S5). As was predicted by the motif analysis, p53 occupancy in MCF10A is significantly higher in lost DARs relatively to gained DARs located in TADs of up- and down-regulated genes (Fig. 6e, f). Noteworthy, p53 binding loci in MCF10A undergo massive loss of chromatin accessibility (56% lost DARs, *p* < 2.2e-16, Supplementary Fig. S6), reinforcing that rewiring of p53 regulatory network contributes to transcriptional reprogramming in G12V cells. The variation in p53 binding was accompanied by a significant decrease in the proportion of canonical p53 binding motif in cell type-specific peaks (from 93 to 69%, *p* < 2.2e-16, proportional test, Supplementary Fig. S7), suggesting that p53 binding in G12V cells is mediated by another transcription factor. Interestingly, the binding motif of AP-1 family of transcription factors is highly enriched in G12V-specific p53 binding loci (24%) relatively to the entire ChIP-seq dataset in these cells (14%) and particularly relative to the MCF10A-specific binding loci (6%), which is close to the background level. This supports a major role

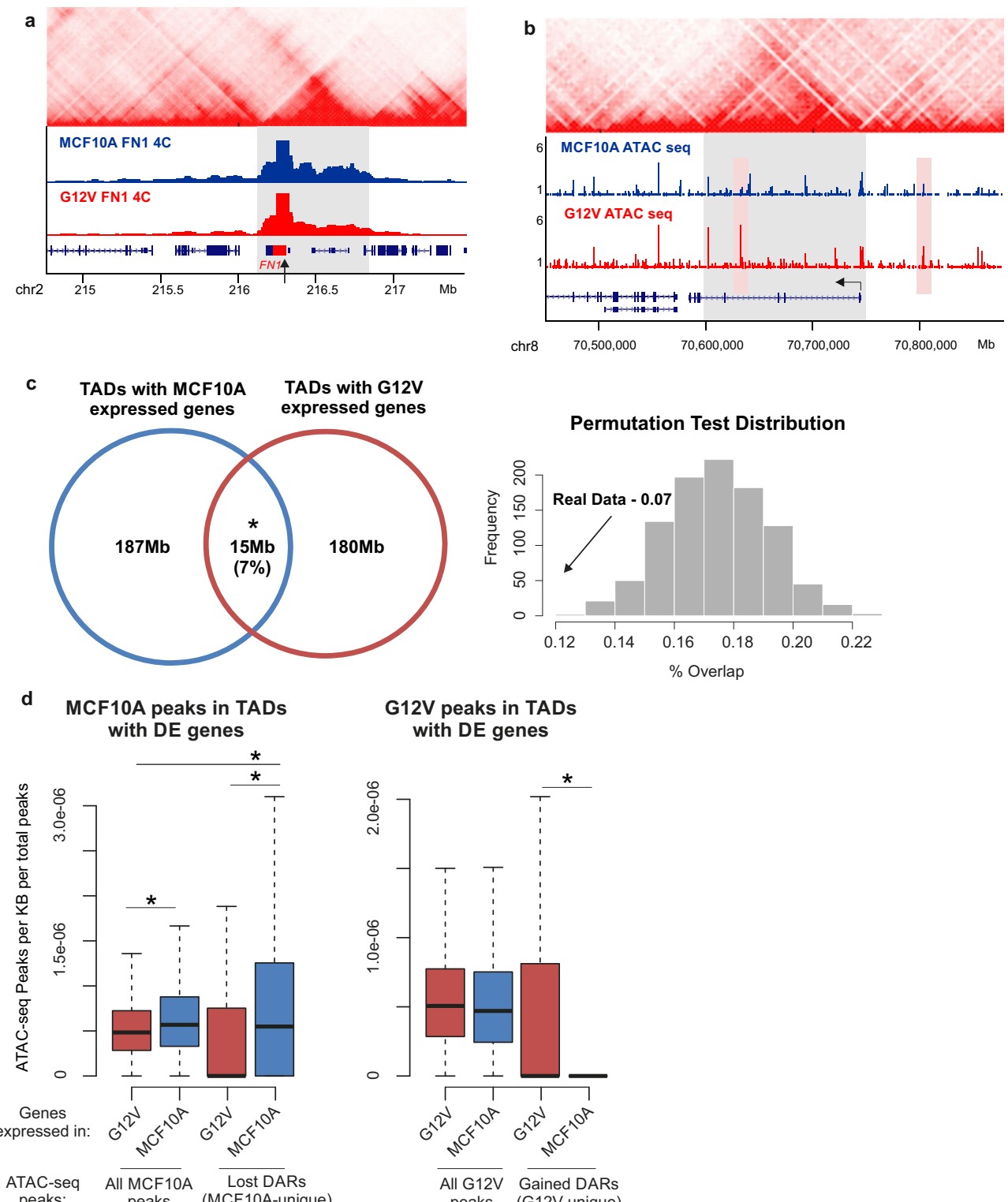

for p53 in the transcriptional reprogramming during the early stages of HRas oncogene-induced transformation. Importantly, this reprogramming by p53 does not occur through changes in protein level or localization, but rather through changes in its chromatin binding.

In order to define how the redistribution of p53 binding is related to disease functions, we first assigned MCF10A-specific and G12V-specific binding sites to their 3D-associated differentially expressed genes. The 84 down-regulated genes associated

with MCF10A-specific p53 binding sites were enriched in disease functions of the formation of cytoskeleton and actin filaments. GREAT analysis assigned to MCF10A-specific binding sites retrieved terms related to development and differentiation functions, in line with the loss of cellular identity and increased plasticity in cancer progression (Supplementary Data 1). The 95 up-regulated genes within TADs decorated with G12V-specific p53 binding sites were enriched in multiple cancer-related functions such as cell invasion, cell viability, and cell proliferation as

**Fig. 4 TADs as the spatial framework of transcriptional regulation. a** Example of 4C-seq profile of the down-regulated *FN1* gene (marked with black arrow) in MCF10A (blue track) and G12V cells (red track) showing there are no changes in FN1 domain (marked with gray box) borders after transformation. Hi-C data from GM12878 is shown on the top[26]. Chromosomal coordinates in Mb of human hg19 genome build are indicated at the bottom. **b** Example for a domain of an up-regulated gene, *SLCO5A1*, (marked with a gray box, TSS marked with black arrow) as defined from Hi-C data (shown on top,[26]) and the ATAC-seq data in MCF10A and G12V cells. Right pink box indicates a unique regulatory site relatively close to the TSS which is outside of the TAD and the left pink box indicates a further away regulatory site that is within the TAD. Chromosomal coordinates in Mb of human hg19 genome build are indicated at the bottom. **c** Left- Venn diagram showing the overlap in Mb between the domains of down-regulated (MCF10A-specific) genes and up-regulated (G12V-specific) genes after HRas transformation. *$p < 0.001$, permutation test. Right- histogram showing permutation test results for the degree of overlap between up-regulated and down-regulated genes in the same TAD. Real overlap (shown with an arrow) is significantly lower than the overlap expected given random distribution. **d** Boxplots showing ATAC-seq peak density in domains of up- (red) or down-regulated genes (blue). Left – MCF10A regulatory sites, right- G12V regulatory sites. *$p < 0.01$, Wilcox test.

well as cellular processes related to G1/S transition and response to DNA damage. Interestingly apoptosis-related functions were not enriched in these analyses (Supplementary Data 2). Thus, genes which may be directly reprogrammed by the redistribution of p53 binding are related to an important subset of p53 functions. Finally, we asked to what extent the cancerous phenotypes of the transformed cells are related to p53. While activation of p53 by Nutllin-3a inhibited cell proliferation of both, MCF10A and G12V cells, this response was initially attenuated in G12V cells, which also maintained their capacity to migrate (Fig. 6h, i). The disorganized growth of G12V cells in 3D cell culture setup is a hallmark of cell transformation (Fig. 6j). p53 activation by mild Nutlin-3a treatment reversed the effect of HRAS overexpression on mammosphere formation, while reduced the size of the mammospheres from MCF10A cells. These results indicate that the redistribution of p53 binding did not eliminate its protective capacity but may have diminished some of its arms.

## Discussion
Cancer development is associated with altered gene expression programs which are key in the acquisition of biological capabilities that drive tumorigenesis. Defining the regulatory networks underlying carcinogenesis associated transcriptional reprogramming holds great promise for identifying transcription factors in this process that may not harbor mutations or change expression patterns. Transforming cells by overexpressing oncogenes is widely used and has been instrumental in understanding the molecular mechanisms involved in malignancy[29]. These cell-based models allow exploring processes occurring in the early stages of oncogene-induced transformation. Using a cell-based mammary epithelial model we find that transformation induced by overexpression of oncogenic HRas is associated with dramatic changes in gene expression. Indeed, the changes we report are characteristic of a carcinogenic transformation process.

Regulatory sites are known to malfunction and thus cause major gene expression alterations related to cancer[30,31]. Several chromatin characteristics are used as a proxy for activity of regulatory elements, including DNA methylation status, nucleosome occupancy, specific histone species and modifications in flanking regions. These were used in several studies to demonstrate major alterations in the DNA regulatory landscape in various cancer cell lines and tumor cells from different cancer types[31–35]. By assaying chromatin accessibility, we demonstrate that even at early stages of transformation, induced by over expression of oncogenic G12V HRas variant, major alterations in the regulatory landscape are evident. Similar modulation of the regulatory landscape was described previously during Ras-dependent oncogenesis in drosophila[36] and using H3K27ac marking following disruption of the ERK signaling pathway in MEF cells[37]. Strikingly, the examination of the distribution of regulatory chromatin between promoters and TSS-distant loci revealed that the latter are dramatically enriched in the varying fraction of accessible

chromatin between transformed and non-transformed cells, illustrating that they have a substantial role in transcriptional reprogramming in oncogenic HRas induced transformation.

In order to interrogate more thoroughly specific transcriptional regulation of differentially expressed genes, it is necessary to assign altered regulatory elements to their target differentially expressed genes. Distance between regulatory elements and their target genes was shown to vary in the range of hundreds of kbs[18], however, regulation of gene expression is constrained by the 3D organization of the genome[38]. TADs are topological domains in the length of hundreds of kbs of high frequency physical association which are considered stable across different cell types and to some extent even different species[26,39,40]. Therefore TADs are considered a framework within which enhancer-promoter dynamic interactions occur[19,38]. Importantly, TADs were shown to act as co-regulated functional units in different processes from long-term differentiation processes[40] to short-term responses to external signals[23,24]. Strikingly also in the process of oncogene-induced transformation we find strong segregation between up-regulated and down-regulated genes among TADs, which strongly supports that TADs comprise the regulatory context within which transcriptional alteration occurs during early stages of transformation. Moreover, we confirm that there are no major changes in TAD boundaries for a number of differentially expressed genes as a results of the transformation process.

Motif enrichment in gained DARs that are associated with differentially expressed genes can infer the related transcriptional regulation activity. The transcriptional regulator CTCF was found to be enriched in specific active regulatory sites in G12V cells despite lack of change in its expression. Mutations in *CTCF* are frequently found in breast tumors[3,41], some of which have been shown to affect its DNA binding. Interestingly, the effect of mutations on CTCF binding was found to be non-uniform, for instance different mutations in its zinc finger 3 domain led to selective inhibition of binding to different targets suggesting that there is a tumor-specific change of function rather than loss of function[41]. Furthermore, CTCF was shown to be elevated in breast cancer cell lines and breast tumors[42], its overexpression was suggested to protect tumor cells from induction of apoptotic cell death[43] while its downregulation in breast cancer cell lines was shown to be associated with reduced cell proliferation[44,45], both effects via transcriptional regulation of target genes. In line with these results we find overall increased chromatin binding of CTCF in G12V cells, although without detectable elevation in its gene expression levels.

P53 is a known tumor suppressor and the *TP53* gene is frequently mutated in breast cancer[3]. We found the p53 binding motif to be strongly enriched in domains of differentially expressed genes in response to HRas oncogenic activity. Moreover, these genes were indeed regulated by p53 as activation of p53 by Nutlin specifically increased their expression level. However, like CTCF, we found that p53 chromatin binding is redistributed while its expression levels are not changed as well as its

**a   Motif enrichment in TADs with UP-regulated genes**

| Motif | P-value | % of Target Sequences with Motif | % of Background Sequences with Motif |
|---|---|---|---|
| ETV1(ETS) | 1.00E-12 | 26.04% | 15.24% |
| Fli1(ETS) | 1.00E-10 | 21.13% | 12.28% |
| CTCF | 1.00E-06 | 5.06% | 1.90% |
| Nur77(NR) | 1.00E-04 | 2.98% | 1.05% |

**Motif enrichment in TADs with DOWN-regulated genes**

| Motif | P-value | % of Target Sequences with Motif | % of Background Sequences with Motif |
|---|---|---|---|
| CTCF | 1.00E-12 | 8.53% | 1.78% |
| Fli1(ETS) | 1.00E-09 | 22.74% | 11.49% |
| ETV1(ETS) | 1.00E-08 | 25.58% | 14.47% |
| BORIS | 1.00E-07 | 8.01% | 2.44% |

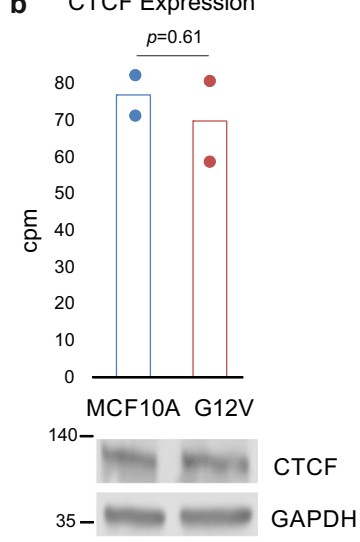

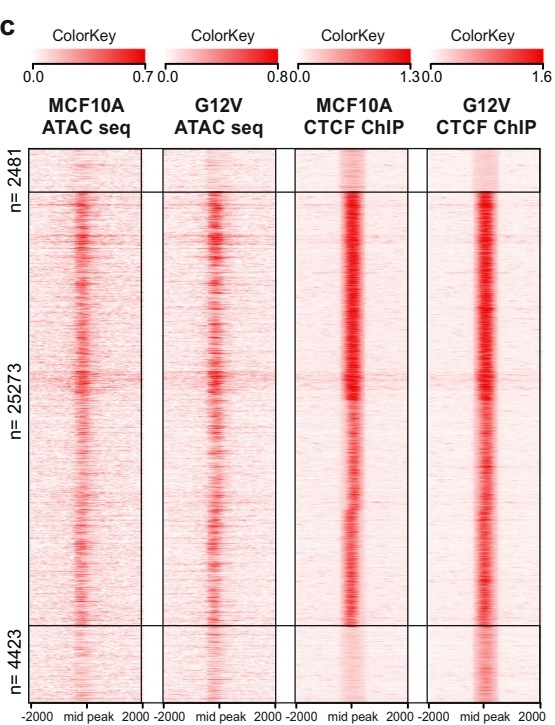

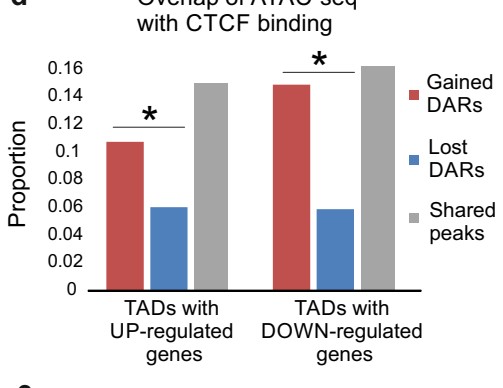

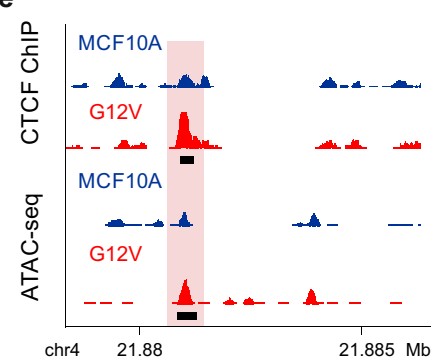

**Fig. 5 CTCF motif and binding sites are enriched in gained DARs. a** Top motifs enriched in gained DARs within TADs of up- or down-regulated genes. Background group is the lost DARs. **b** Expression levels of *CTCF* in MCF10A (blue) and G12V (red) cells determined by RNA-seq. Data points and average (bar graph) are presented. *p*-value determined by DESeq2 - Wald test corrected for multiple testing using the Benjamini and Hochberg method. CTCF protein levels in MCF10A and G12V cells as measured by western blot analysis (bottom). Representative result of at least two biological repeats is shown. **c** Heatmap displaying k-means clustering of CTCF ChIP-Seq data in the two cell types. The ATAC-seq data were arranged to match the order of loci found by clustering CTCF ChIP-Seq. Four kb around the ChIP-seq peaks are displayed. **d** % of gained (blue) and lost (red) DARs overlapping CTCF binding sites in TADs of up- and down-regulated genes. *$p < 0.001$, proportional test. **e** Example for a gained DAR that overlaps with gained CTCF binding site (pink box). Black boxes indicate peaks, the black line indicates CTCF motif. Chromosomal coordinates in Mb of human hg19 genome build are indicated at the bottom.

localization in the cell. The involvement of p53 in cancers is associated with its DNA binding and transcriptional regulation activity[46]. Our results suggest that distinct genomic p53 binding patterns reported in cancer cells[47–51] may be important for the development of capabilities during the progression of cancer. Given that p53 binding profile is cancer-specific, it is unlikely that p53 alone reprograms the chromatin landscape[52]. Interestingly,

redistribution of p53 binding is accompanied by a decrease in the proportion of its binding motif and an increase in AP-1 binding motif. This suggests a possible role for members of the AP-1 family of transcription factors in modulating p53 activity in the transformed cells by direct association with p53 binding sites[53]. Functionally, this modulation may relate to transcriptional suppression, as p53 represses target genes of TFF2 via the AP-1

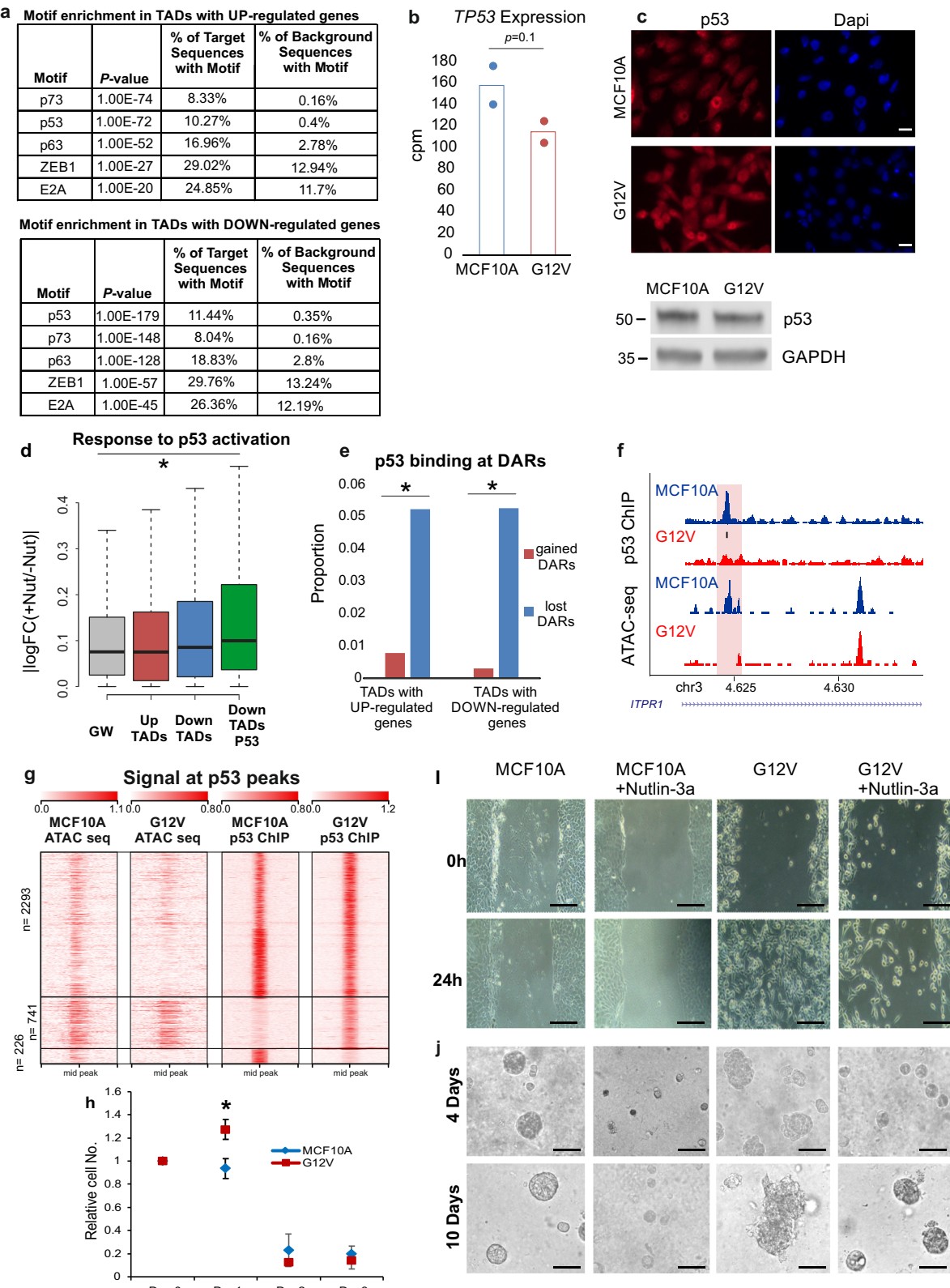

motif[54]. In addition, *ATF3* and *FOSL1* subunits of the AP-1 transcription factor complex that were up-regulated in G12V cells, were shown to have regulatory interactions with p53[55,56].

The 3D conformation of the genome is the framework within which transcriptional regulation takes place. Here we take advantage of this framework together with DNA accessibility as proxy for active regulatory elements, to infer from changes in gene expression

upon HRas oncogene overexpression in normal breast epithelial cells, on regulatory networks that are dis-functioning at the first stages of oncogene-induced carcinogenesis. Chromosome structure Hi-C data allowed combining regulatory loci that change their activity with their distant transcriptionally responsive gene-targets in their biologically relevant 3D context on a genome-wide scale. This focused approach coupled with motif discovery analysis

**Fig. 6 Redistribution of p53 binding underlies genetic reprogramming and cancer phenotypes. a** Top motifs enriched in lost (MCF10A-unique) DARs within TADs of up- or down-regulated genes. Background group is gained DARs. **b** RNA levels of *TP53* in MCF10A (blue) and G12V (red) cells determined by RNA-seq. Data points and average (bar graph) are presented. p-value determined by DESeq2. **c** Immunostaining of p53 (top panel) in MCF10A and G12V cells. Bar = 20μm. p53 protein levels in MCF10A and G12V cells as measured by western blot analysis (bottom). Representative results of at least two biological repeats are shown. **d** Variation in RNA levels |log2FC| following 4 h Nutlin-3a treatment in MCF10A cells. Gray—all expressed genes, red –up-regulated genes after HRas transformation, blue—down-regulated genes after HRas transformation and green—down-regulated genes that have lost regulatory sites with a p53 motif in their domain. *p < 0.001, Wilcox test. **e** % of gained (red) and lost (blue) DARs overlapping p53 binding sites from MCF10A and G12V ChIP-seq in up and down TADs. *p < 0.001, proportional test. **f** Example for a lost DAR that overlaps with lost p53 binding site (pink box). Blackline indicates p53 motif. Chromosomal coordinates in Mb of human hg19 genome build are presented. **g** Heatmap displaying k-means clustering of p53 ChIP-Seq data in the two cell types. The ATAC-seq data were arranged to match the order of loci found by clustering p53 ChIP-Seq. Four kb around the ChIP-seq peaks are displayed. **h** Cell proliferation with 5 μM Nutlin-3a was measured by XTT assay. The relative number of cells, compared to day 0 is presented for MCF10A (blue line) and G12V cells (red line). *p = 0.0002 T-test **(i, j)** Representative images showing the migration capability and growth pattern of MCF10A and G12V cells with or without Nutlin-3a. Scale bar (i) =400 μm, Scale bar (j) =100 μm.

revealed two regulatory factors, namely CTCF and p53, that regulate transcriptional variations associated with HRas oncogenic cellular transformation. Noteworthy, these factors carry out this modular activity, while their RNA, protein levels, and subcellular localization remain invariable, therefore could have not been identified by differential expression analysis. Our ATAC-seq and ChIP-seq data support that this effect is occurring through changes in DNA binding patterns. Thus the combination of differential expression analysis, and DNA accessibility using the framework of the 3D organization of the genome is a powerful tool to identify in an unbiased manner regulatory pathways that orchestrate transcriptional reprogramming during the early stages of cancer development.

## Methods

**Cells and treatments**. MCF10A cells were grown as previously described (Debnath et al.) in DMEM media (Biological Industries) supplemented with 100 ng/ml cholera toxin (Sigma), 20 ng/ml epidermal growth factor (EGF, Peprotech), 0.01 mg/ml insulin (Sigma), 500 ng/ml hydrocortisone (Sigma), 1% penicillin-streptomycin (Biological Industries), 5% horse serum (Biological Industries). Introduction of the G12V HRas oncogene was done via lentiviral transduction along with a GFP expression vector in order to track transduction efficiency. After 72 h lentiviral transduction hygromycin was added to the media in order to select for cells carrying the G12V HRas expression vector. Nutlin treatment was done by adding 10 μM Nutlin-3A (Sigma) to the media for the periods of time indicated.

**Lentiviruses**. Lentiviral particles were prepared by cotransfecting either the G12V HRas overexpression vector pWZL hygro HRas V12 (addgene #18749) or a GFP expression vector with packaging vectors (CMVΔR8.91, CMV-VSV-G) into 293 T cells using Mirus TransLTi (Mirus Bio LLC, Madison, WI) according to the manufacturer's instructions. Medium containing viral particles was collected 2 and 3 days post-transfection.

**Proliferation assay**. Cell proliferation was measured for 3days using an XTT based cell proliferation kit (Biological industries), according to the manufacturer's instructions.

**Colony formation assay**. In total 100 cells per well were seeded in 6 well-plates in triplicates. After two weeks, cells were fixed and stained with Giemsa stain. Colonies larger than 5 mm were counted.

**Soft agar assay**. A total of $2 \times 10^4$ cells were seeded in MCF10A standard media containing 0.3% 2-Hydroxyethyl Agarose on top of a solidified layer of MCF10A media containing 0.6% 2-Hydroxyethyl Agarose. Plates were incubated until colonies were visible by the naked eye at which point they were counted.

**Anchorage-independent cell death assay**. In total $5 \times 10^3$ cells were plated onto poly-HEMA coated 12-well plates to prohibit attachment. After 4 days in suspension, cells were collected from the wells and live cells were counted manually using Trypan blue exclusion.

**Matrigel invasion assay**. Cells were placed in the upper chamber of a Boyden migration chamber. The upper and lower chambers were separated by matrigel coated PVP-free, 8-mm pore size polycarbonate filters (Costar Scientific). EGF containing conditioned medium of 3T3 fibroblasts was placed in the lower chamber. After overnight incubation, the filters were fixed and stained with Diff-

Quick System (Dade Behring, Inc.) and cells on the lower surface were counted. Each assay was done in triplicates.

**In vivo tumorigenesis**. About $5 \times 10^6$ cells were injected s.c. on both dorsal sides of 6 weeks old non-obese diabetic/severe combined immunodeficient (NOD/SCID) mice (4/group) and tumor formation and size were followed for 8 weeks, after which animals were killed. All studies with mice were approved by the Institutional Animal Care and Use Committee at the Hebrew University of Jerusalem.

**RNA extraction, cDNA synthesis, and qPCR**. Total RNA was extracted from the cultured cells using Quick-RNA MiniPrep kit (ZYMO RESEARCH) following manufacturer's instructions, reverse transcribed using Quanta Bioscience qScript cDNA synthesis kit (95047-100) following manufacturer's instructions. cDNA was measured by real-time PCR (Bio-Rad S1000) using sybr green mix (Bio-Rad) with primers spanning exon- intron junctions (Supplementary table S1) and normalized to GAPDH transcript. Results show average and SD of three replicates.

**Wound healing assay**. About $25 \times 10^4$ cells/well were seeded in 12 well plate wells to form a 100% confluent layer. One day after the layer was wounded using the 10 μl pipet tip and monitored for wound healing at 0 h and 24 h in minimal growth factor medium.

**3D culture assay**. Matrigel was incubated on ice overnight, then 3000 cells were seeded on a solidified layer of growth factor reduced Matrigel measuring approximately 1–2 mm in thickness. The cells were grown in an assay medium containing 5 ng/ml EGF and 2% Matrigel then incubated in $CO_2$ incubator at 37 °C.

**RNA-seq**. Total RNA was extracted from cells using RNA purification kit (GeneAll) according to the manufacturer's instructions. RNA quality was measured on a Bioanalyzer (Agilent) and the only RNA with RIN score >9 was used for library preparation. Messenger RNA (mRNA) was enriched from 1 μg of total RNA by Poly(A) mRNA Magnetic Isolation Module (New England Biolabs) according to the manufacturer's instructions. cDNA libraries were constructed using the NEBNext Ultra RNA Library Prep Kit (New England Biolabs) following the manufacturer's protocol. Library concentration was measured by DNA High Sensitivity Kit (Invitrogen) on a Qubit fluorometer (Invitrogen). Library quality and fragment sizes were assessed on a Bioanalyzer (Agilent) or on a Tape station (Agilent). RNA-Seq libraries from at least two biological replicas for each condition were sequenced on Illumina Hi-seq 2000 platform.

**ATAC-seq**. ATAC-seq was performed as previously described[57]. Briefly, cells were lysed in NLB buffer (10 mM Tris-HCl pH 7.5, 10 mM NaCl, 3 mM MgCl2, 0.05% NP-40 and protease inhibitors (Sigma, P2714). Transposition reaction was performed on $10^5$ nuclei using 5 μl of Nextera TDEI enzyme (Illumina, FC-121-1030) for 30 min at 37 °C. DNA was then purified by Expin PCR SV (GeneAll, 103-102) and the library was amplified using NEBNext High-Fidelity 2× PCR Master Mix (New England Biolabs, M0541). The libraries were size-selected by a gel-free double-sided size-selection using Agencourt AMPure XP beads (Beckman, 63881), at 0.5X and 1.2X. Library concentration was measured by DNA High Sensitivity Kit (Invitrogen) on a Qubit fluorometer (Invitrogen). Library quality and fragment sizes were assessed on a Bioanalyzer (Agilent). ATAC-Seq libraries from two biological replicas for each condition were sequenced on Illumina Hi-seq 2000 platform.

**ChIP-seq**. Cells were cross-linked for 10 min at 37 °C in 1% formaldehyde followed by quenching with 125 mM glycine for 10 min. for CTCF, crosslinked cells were first lysed in cell lysis buffer (10 mM Hepes pH7.5, 85 mM KCl, 1 mM EDTA, 1% NP-40) supplemented with protease inhibitors, resuspended in RIPA buffer (10 mM Tris-HCl, pH 7.6, 1 mM EDTA, 0.1% SDS, 0.1% NaDeoxycholate, 1% Triton X100) supplemented with protease inhibitors, and sonicated for 40 cycles of 30 s

ON and 30 s OFF (Bioruptor sonicator, Diagenode). Cleared chromatin was incubated overnight with 10 μg α-CTCF (Millipore 07-729) or 5 μg α-p53 (DO-1, Santa-Cruz) and additional 2 hours with 40ul protein A/G magnetic beads (ChIP grade, Pierce). Complexes were washed twice with RIPA buffer, twice with RIPA buffer supplemented with 300 mM NaCl, twice with LiCl buffer (10 mM Tris 7.5, 1 mM EDTA, 0.25 M LiCl, 0.5% NP40, 0.5% NaDeoxycholate), once with TE buffer (10 mM Tris, 1 mM EDTA, pH8) supplemented with 0.2% triton and once in TE buffer. For p53, cells were treated with 10uM Nutlin-3a (Sigma) for 4 h before fixation. Fixed cells were lysed with SDS Lysis Buffer (1% SDS, 50 mM Tris pH 8.1, 10 mM EDTA) supplemented with protease inhibitor and sonicated for 560 s (ME220 sonicator, Covaris). Cleared chromatin was diluted 1:10 with dilution Buffer (16.7 mM Tris-HCl 8.1; 1.2 mM EDTA; 167 mM NaCl; 1.1% Triton) and incubated overnight with 5 μg α-p53 (DO-1, Santa-Cruz) bound to magnetic beads (Dynabeads Protein A). Chromatin was washed with low salt buffer (20 mM Tris-HCl 8.1; 2 mM EDTA; 150 mM NaCl; 1% triton; 0.1% SDS), high salt buffer (20 mM Tris-HCl 8.1; 2 mM EDTA; 500 mM NaCl; 1% triton; 0.1% SDS), LiCl buffer (10 mM Tris-HCl 8.1; 1 mM EDTA; 1% NP-40; 250 mM LiCl) at 4 °C, and twice with TE (10 mM Tris-HCl 8.1; 1 mM EDTA) at room temp. Complexes were eluted with Elution buffer (10 mM Tris-HCl 8.1; 1 mM EDTA; 200 mM NaCl; 1% SDS). For both antibodies, crosslinks were reversed with 1 mg/mL Proteinase K overnight at 65 °C. Purified DNA was used to prepare sequencing libraries using NEBNext UltraII DNA Library Prep Kit (New England Biolabs). Library concentration was measured by DNA High Sensitivity Kit (Invitrogen) on a Qubit fluorometer (Invitrogen). Library quality and fragment sizes were assessed on a Bioanalyzer (Agilent). ChIP-seq libraries from two biological replicas for each condition were sequenced on Illumina Hi-seq 2000 platform.

**4C-seq**. 4 C was performed as previously described[58,59]. Cells were fixed with 2% formaldehyde for 10 min, cross-linked overnight with an excess of HindIII enzyme (New England Biolabs) and then DNA ends were ligated under dilute conditions that favor junctions between cross-linked DNA fragments. The ligation junctions were then circularized by digestion with Csp6I (Thermo Scientific). Chromosomal contacts with the baits were amplified with inverse PCR primers (Supplementary Table S2) using Platinum Taq DNA Polymerase (Life-Technologies). 4C-seq libraries were sequenced on Illumina Hi-seq 2000 platform.

**Immunostaining**. Cells were fixed with 3.7% formaldehyde, permeabilized with 0.1% triton, and blocked with 2% BSA. p53 was detected using a mouse monoclonal antibody (SC-126, Santa Cruz Biotechnology) followed by a Cy3 conjugated anti-mouse antibody (Jackson Immunoresearch Laboratories). Cells were stained with Dapi and viewed on an Axioimager fluorescent microscope (Zeiss).

**Western blot analysis**. Cells were lysed directly in sample buffer and protein extracts were separated on a 4–20% polyacrylamide gel (Bio-Rad). Blotted nitrocellulose membranes were incubated with primary antibodies (mouse α-p53, SC-126, Santa Cruz Biotechnology; rabbit α-CTCF, Millipore 07-729; rabbit α-GAPDH, cst-2118, Cell Signaling Technology) followed by detection with HRP conjugated secondary antibodies (Jackson Immunoresearch Laboratories). ECL was performed using EZ-ECL kit (Biological Industries) and imaged on an ImageQuant chemiluminescence camera (GE Healthcare).

**Statistics and reproducibility**. Two biological replicates were used for each genome-wide (GW) analysis. To assess the reproducibility of ATAC seq and ChIP-seq replicas correlation was calculated using sampling 10 M reads per replica, calling peaks, and calculating the correlation between the merged peaks. The correlation was between 0.79 and 0.9. The reproducibility of the RNA seq was evaluated using PCA and hierarchical clustering. Statistical tests were conducted using the R programing language.

**RNA seq analysis**. The alignment was done using TopHat[60]. Reads count on transcripts was done using HTSeq[61]. Expression and differential analysis was done using "edgeR"[62] that assigns to each gene a false discovery rate (FDR) value and calculates the log Fold Change (logFC) between the two conditions. Genes with FDR < 0.05 and logFC greater than |0.5| were considered differentially expressed. Functional enrichment was done using IPA (Qiagen,[63]).

**ATAC seq and ChIP-seq analysis**. The alignment was done using Bowtie[64], allowing only unique reads to be considered. Since the replicas were well correlated (Supplementary Fig. S2), reads from both replicas of each condition were combined for further analysis. For detecting regions of local read enrichment (peaks), MACS algorithm[65] was applied with default parameters for ChIP-seq. For ATAC-seq, read start site was adjusted to represent the center of the transposon binding event as described[66] and the following parameters for MACS were applied – --tsize=51 --nomodel --shiftsize=75 --llocal=25000 -p 1e-04. Differentially accessible loci between MCF10A and G12V cells were detected by MACS algorithm, using one sample as the background of the other. Only peaks that were both detected by MACS analysis in either sample and in the comparative MACS analysis were considered as unique peaks and were used for further analysis.

For calculating the distribution of ATAC-seq peaks relatively to gene features, refseq annotations were used for TSS coordinates of each gene, and the longest transcript as gene body. Overlap of at least 1 bp between ATAC-seq peak with gene promoters (1000 bp upstream and 500 bp downstream to the TSS) or with distant regulatory regions (100 kb upstream and downstream to the TSS of each gene) was counted. Regulatory sites that did not overlap one of the two features were considered as far regulatory sites. Overlap analysis was performed by intersectBed from Bedtools[67]. Each gene or peak was counted once.

**4C-seq analysis**. Reads were sorted, according to their barcodes, to different fastq files for each bait and condition and aligned to the human (hg19) genome using BOWTIE[64]. Reads were then counted for each HindIII site. For domains detection, the number of reads on each HindIII site was counted in sliding windows of 50Kb with 25Kb steps. Hi-C heatmaps are from the 3D genome browser[68].

**Association of genes and TADs**. For genome-wide assignment of the gene to topologically associated domains (TADs), TADs from high-resolution Hi-C data of 7 human cell lines[26] were merged to one dataset and the smallest overlapping TAD was assigned for each gene. "Up" or "down" TADs were declared according to their containing up- or down-regulated genes after transformation by G12V HRas. Overlap between TADs, regulatory sites, and genes was calculated by intersectBed from Bedtools[67]. For the GO analysis presented in Supplementary Data 1, cell-type-specific expressed genes within "Up" or "down" TADs that contain cell-type-specific ChIP peaks were used for IPA analysis. For GREAT analysis[69], differential factor binding loci were retrieved by EdgeR package from R (p.adj < 0.05) using merged peaks from all replicas as input.

**Motif discovery analysis**. To uncover DNA binding motifs of possible transcription factors (TFs) which are associated with regulatory loci, HOMER "findMotifsGenome.pl" program[70] with default parameters was applied for a 150 bp window with the highest read count within each peak (summit) that was defined using the custom made script in R. The enrichment of motifs in regulatory sites from MCF10A was measured against G12V cells and vice versa.

**Reporting summary**. Further information on experimental design is available in the link to this paper.

## Data availability

RNA sequencing, ATAC sequencing, ChIP-sequencing, and 4 C sequencing data have been deposited at GEO, accession number GSE140254. Source data can be found in Supplementary Data 3. All other data can be obtained from the corresponding author on a reasonable request.

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

## Acknowledgments

This work is supported by the Israel Science Foundation (grant 748/14), Marie Curie Integration grant (CIG)- FP7-PEOPLE-20013-CIG-618763 and I-CORE Program of the Planning and Budgeting Committee and The Israel Science Foundation grant no. 41/11. We thank the BIU imaging facility for helping with microscopy.

## Author contributions

M.S., B.Z.A., M.T., O.L., A.C., and Y.T. performed experiments, M.S., A.S.P., L.C., T.K., Z.S., and O.H. analyzed data. M.S., A.S.P., Z.S., and O.H. designed experiments. M.S., A. S.P., and O.H. wrote the manuscript.

## Competing interests
The authors declare no competing interests.
