## [Peer Review File · Communications Biology]

Reviewers' Comments:

Reviewer #1:

None

Reviewer #2:

Remarks to the Author:

Transcriptional dysregulation is a near-universal feature of carcinogenesis, but the molecular details underlying cancer-promoting transcriptional reprogramming are, in most cases, poorly understood. Here, the authors describe large-scale changes in chromatin accessibility and gene expression in cells expressing oncogenic HRas-G12V, focusing on redistribution of the transcriptional regulators CTCF and P53 as a potential driving force for the chromatin and transcription alterations observed. The authors show that differential chromatin accessibility occurs within stable TADs at tested genes using 4C, and that differentially accessible regions harbor motifs for a number of interesting factors, including CTCF and P53 (as well as oncogenic ETS factors). The results are intriguing, suggesting roles for two well-known transcriptional regulators in the transcriptional rewiring that accompanies G12V transformation, and should stimulate further mechanistic investigation of carcinogenic transcriptional reprogramming. However, the presentation of the data is rather haphazard, with confusing terminology as I elaborate on below. Also, the analysis of the ChIP-seq data is rather narrow, focusing only on regions specific to one cell type or the other – I think a broader analysis of the data is needed.

Specific comments:

Reference 10 (The ENCODE Nature paper) is provided as a reference for the capacity of histone modifications to demarcate active enhancers. However, several publications prior to this one showed that H3K4me1/H3K27ac is a signature of active enhancers: Creighton et al, PNAS 2010 (PMID 21106759); Rada-Iglesias et al, Nature 2011 (PMID 21160473); Zentner et al, Genome Res 2011 (PMID 21632746).

Statistical analysis should be included for the phenotyping data presented in Fig. 1 and 6.

The authors interchangeably use gained and lost DARs with G12V-unique and MCF10A-unique. I found this confusing throughout, with one example being Fig. 5C, where the label uses the G12V/MCF10A nomenclature while the legend refers to gained and lost DARs. Adding to the confusion is the use of "parental" to refer to non-transformed MCF10A. This terminology should be made consistent throughout.

Fig. 5C refers to "CTCT" binding sites, which I assume should be "CTCF."

I'd like to see some expansion of the analysis of gained and lost CTCF and P53 binding sites. I understand that the authors performed differential peak calling using MACS, but it would be interesting to see heatmaps generated using composite lists of peaks regardless of whether they are common or unique to MCF10A or G12V cells. This would give an idea of how complete the gain or loss of signal at these regions is. As it stands, for CTCF, the single example shown in track form in Fig. 5D suggests that there is some CTCF binding at this region in MCF10A cells. For P53, it appears that it may be a complete loss of binding. I think that heatmapping the ChIP-seq and ATAC-seq data in this manner would give the reader a more complete view of the data.

In Fig. 6E, it is shown that lost DARs are enriched in P53 binding; that is, MCF10A-specific accessible sites are P53-bound (I assume this is ChIP-seq data from MCF10A cells). In the corresponding text, it is stated that this result confirms the direct role of P53 in reprogramming transcription in G12V cells. It seems more that this suggests a suppression of transcriptional rewiring, as these sites are lost in G12V cells (at least in the 1 example shown in Fig. 6F). Again, it would be helpful to see how widespread the loss of P53 binding really is at lost DARs.

Have the authors deposited their data in GEO? If so, an accession number and reviewer token should be provided for confirmation.

The authors should make any custom scripts used for analysis available on Github or another repository.

Reviewer #3:

Remarks to the Author:

The authors over-expressed a mutated HRas (G12V) in an epithelial cell line MCF10A and explored the potential mechanism that drives the transcriptional reprogramming. They compared gene expression (RNA-Seq) and chromatin accessibility (ATAC-Seq) between the two lines (MCF10A vs G12V). The expression changes match phenotype changes induced by HRas, relating upregulated genes to invasion and downregulated genes to apoptosis (Fig 1 & Fig 2). The ATAC-Seq data analysis revealed changes in accessibility at tens of thousands of regulatory sites (differentially accessible regions; DARs), correlated well with expression changes (Fig 3). To predict target genes for distal regulatory sites, the authors first showed that the up-regulated genes and the down-regulated genes do not likely reside in the same topologically associating domain (TAD). They further showed that cell-type specific DARs are enriched within TADs containing cell-type specific expressed genes (Fig 4). They author then narrowed down to DARs that are associated with differential expressed genes within the same TADs, carried out motif enrichment analysis, identified potential regulators including CTCF (Fig 5) and P53 (Fig 6), and confirmed the bindings by ChIP-Seq analysis. The two transcription factors exhibited a redistribution of binding as induced by HRas without a change in protein level. The authors also confirmed the transcription response induced by P53 (Fig 6). Many of the observations are predictable based on literature (especially for Figs 2,3 and 4). The findings on Figure 5 and 6 require further evidence to understand the mechanism regarding the redistribution of TF binding induced by HRas.

- 1) The authors used TAD to connect DAR to differentially expressed (DE) genes. Details of the method are missing. For example, how would the author assign the target(s) of a DAR, if a TAD contains multiple DE genes and multiple DARs?
- 2) The benefit of using TAD to choose subsets of DARs for motif analysis is unclear. Would applying motif analysis to non-promoter DARs without considering TAD yield similar hits to those shown in Fig 5A and Fig 6A. In addition, as separating DARs based on upregulated genes and downregulated genes gave similar motifs with comparable % of target sequences (Fig 5A and Fig 6A), the benefit of considering expression changes seems marginal.
- 3) The author showed that cell type specific DARs are enriched within TADs containing cell-type specific expressed genes. However, the data presentation for Figure 4D is confusing. It is difficult to figure out the steps of the data analysis that has led to Figure 4D, even though the conclusion fits expectation.
- 4) The data analysis for regulator inference in Fig 2E is not clear. Did the author use differentially expressed genes as an input or use a pre-ranked list of all genes? I am surprised to see that LPS (lipid) and dexamethssone (a drug) were predicted as two regulators.
- 5) Figure 5C: the author need to include DARs shared by G12V and MCF10A as a control.
- 6) The authors stated a global redistribution of CTCF binding but only included a specific example (Figure 5D) without detailed global analysis. They showed that the number of CTCF peaks increased from 22K to 31K from MCF10A to C12V. But the number of peaks would depend on sequence depth. A differential peak analysis similar to Figs 3A and 3C will help to clarify (also

apply to the P53 ChIP-Seq data analysis).

7) Fig 5D and 6F: are the two regions functional remarkable. Globally, what are the functions for all the differential CTCF binding and P53 binding sites. A GREAT analysis (<http://great.stanford.edu/public/html/>) would give some hints.

8) The authors observed a redistribution of CTCF or P53 binding but did not investigate the mechanisms. From motif analysis, the authors hypothesized that TFs recognizing AP1 motif could involve in the redistribution of P53 binding. I would suggest the author to verify the hypothesis by doing a CRISPR KO at an AP1 site and measured the binding change of P53 on the site in G12V by ChIP-PCR.

9) Some minors: some of the labeling of the subpanels in Fig 6 are not correct; I assumed Fig 6D to be Figure 6B. Figure 5D: scale on the y-axis is missing. GSE# is missing.

We sincerely thank the reviewers and the editor for their relevant and useful comments for our manuscript “Genomic Retargeting of p53 and CTCF in Oncogenesis” which have helped improve its quality. We have addressed the points raised in the review, and rewritten sections of the manuscript accordingly. Below, we replicate the reviewer’s comments (black text) and provide our point-by-point responses inline (red text):

Reviewer #1

COMMSBIO-19-1334. Genomic Retargeting of p53 and CTCF in Oncogenesis

This manuscript aims to uncover the roles of chromatin state and transcription factor (TF) binding in cancer development. The authors focused on the initial stage of oncogene-induced carcinogenic transformation and studied the changes of gene expression and chromatin accessibility to reveal the underlying regulatory network[s]. Overall, this is an interesting paper with a lot of data. The tumorigenic potential of the G12V mutation in MCF10A is well characterized. The genomic analyses (RNA-seq, ATACseq, 4C-seq and ChIP-seq) are pretty comprehensive. However, there are several issues that need to be clarified.

In particular, based on their results (motif analyses and ChIP-seq assays), the authors concluded that two TFs, p53 and CTCF, are “major determinants of transcriptional reprogramming at early stages of HRas-induced transformation” (Abstract). I am not convinced, however, that the real data actually support this conclusion. Throughout the paper, the authors tend to infer ‘exceptional significance’ of the two factors, p53 and CTCF, even when it is unsubstantiated. (Specific examples are given below.)

In my opinion, the authors need to be more cautious. The text has to be GLOBALLY revised (see below).

1. Pg. 4. Results: A single mutation at amino acid 12 (G12V) was introduced into MCF10A cells. It remains unclear why and how the authors made the mutation at this position. They need to provide the evidence about the importance of this amino acid. Was the mutation generated by CRISP/CAS9?

One advantage of the model system used in this study is the transformation of a non-tumorigenic immortalized epithelial cell line (MCF10A) with a single oncogene, G12V HRas. As outlined in the materials and methods, we used the lentiviral vector “pWZL hygro HRas V12” from addgene (#18749) that carries the G12V mutation. This mutation locks HRAS in its active form leading to its constitutive activation.

2. Pg. 9. Fig 3 legend: (C) the promoter proximal (+500 bp, -1000 bp from TSS, **red**) – in fact, promoters are shown in “**blue**”, whereas the mid-range is shown in “**red**” color.

We thank the reviewer for identifying this error. This was corrected.

3. Pg. 10, Line 1: “... The data points to **enhancers** as key elements in regulation of transcriptional reprogramming ...”. It is unclear what data points to enhancers. On Pg. 8, the authors wrote: “...indicating that the major changes in regulatory elements are at gene-distant regions (Fig. 3C)”. Does it mean that all gene-distance regions are considered to be “enhancers”?

Indeed, we referred to TSS-distant regulatory elements (which were called based on their accessibility), as enhancers. We do not intend to make a claim that all gene-distance regions are considered to be “enhancers”. We therefore have changed the term “enhancers” in Pg. 10 to “TSS-distant elements” and the sentence on Pg. 8 has been altered to: “... indicating that the major changes in chromatin accessibility are at gene-distant regions”.

Notably, analyzing published MCF10A H3K4me1 enhancer mark (Choe et al., PMID: 22285863) shows that in agreement with previous studies (for example, Birney et al., PMID: 17571346; Siersbæk et al., PMID: 21427703), 66% gene-distant accessible loci in MCF10A cells are decorated with H3K4me1.

4. Pg. 10, middle: “As shown in Fig. 4B, regulatory elements can be found close to a gene’s transcription start site, however not in the same topological domain, while a further away regulatory element is found in the same topological domain which is more

biologically relevant. Thus using TADs as the spatial platform to connect between genes and their regulatory elements is a robust and biologically relevant approach.”

== First, you write: “the same topological domain [...] is more biologically relevant.”

(Why is it more relevant? Is it an assumption or a conclusion? You don’t explain that.)

Next, you repeat practically the same text and conclude that “using TADs [...] is a robust and biologically relevant approach.” I am afraid I don’t see the logic here.

We regret lack of clarity. We have modified this part to:” ...as a regulatory organizational framework, rather than genomic proximity, to link between TSS and regulatory elements (Fig. 4B).”

5. Pg. 11. The Hi-C map of GM12878 cells is shown on the top in Figs. 4A-B. GM12878 is a normal cell line. I would suggest showing a 3D map of cancer cells, e.g., K562 (Dixon et al. 2018 Nat Genet.)

The tumorigenicity of the cell is not expected to affect TAD locations, which are overall conserved across cell types and conditions. Moreover, the genomic instability of K562 cells may hamper the Hi-C analysis, while MCF10A is a rather genetically stable cell line.

Intra-TAD chromosomal associations display higher variance among different cell types, which may affect the TAD “strength” measured by Hi-C. Given that the capability to delineate TAD structures depends also on the complexity of the Hi-C library (number of paired-end ligation junctions), we presented in Fig. 4 deeply sequenced, high resolution Hi-C data from GM12878 cells.

To illustrate these points, we overlaid Hi-C map from K562 cells (932 million pairs, Rao et al., 2014) on the map from GM12878 cells (4.9 billion pairs) from Fig. 4 (below). Notably, in the global analysis we assigned to each TSS the smallest TAD across several high resolution Hi-C datasets.

6. Pg. 11, Fig. 4C: The sub-titles are confusing (or even wrong). Compare with Fig. 4C legend: “Venn diagram showing the overlap in bp between the domains of up-regulated genes and down-regulated genes after HRas transformation.”

We have changed the sub-titles in Fig. 4C from MCF10A/G12V-expressed to MCF10A/G12V-specific genes. Accordingly, we have revised the legend to “(C) –Left-Venn diagram showing the overlap in Mb between the domains of down-regulated (MCF10A-specific) genes and up-regulated (G12V-specific) genes after HRas transformation”.

7. Fig. 2D: Top five significantly affected pathways according to the DE genes. Fig. 2E: Top ten upstream regulators... “p53 signaling” and TP53 are the fifth ones in the corresponding lists (not the first ones). What is the most compelling evidence that p53 is really exceptional? == I don’t see why the authors claim that p53 and CTCF are the “major determinants [or key regulatory factors] of transcriptional reprogramming at early stages of HRas-induced transformation” (Abstract; page 4, just before “RESULTS” and page 22, before “REFERENCES”)

8. Related to the above. Figs. 5A and 6A (UP-regulated genes): CTCF and p53 are not the factors with the lowest P values. Why do you choose them as the “major determinants” of transcriptional reprogramming? If you insist that the two factors, p53 and CTCF, are indeed ‘exceptionally significant’ you have to present VERY STRONG and CLEARLY FORMULATED evidence for that.

Indeed, transcription programs are regulated by an orchestra of TF operating in various hierarchical and sequential manners. p53 was especially relevant since its pathway was highly enriched (although not the highest) in the IPA analysis as well as its motif being significantly enriched in lost DARs associated with differentially expressed genes. As for CTCF, it was amongst the highest enriched motifs in gained DARs associated with differentially expressed genes and we indeed refer to other enriched motifs, namely of ets factors, which are known to be involved in breast cancer. CTCF in this case was interesting, as its expression does not differ between MCF10A and G12V cells. We agree with the reviewer that despite the high enrichment of p53 and CTCF binding motifs in the differential accessible loci and global redistribution of their genomic binding, their impact on gene expression relative to other factors is yet to be determined. Following the reviewer’s comment, we removed the words “major” and “key” to tone down such statements across the manuscript.

9. Pg. 15: “we confirmed that TP53 gene is not mutated in G12V HRas expressing cells (data not shown)...” The authors should show this data because it is critical. The authors should also demonstrate that there is no mutation in the CTCF gene.

The integrity of TP53 and CTCF was verified by inspecting aligned reads from RNA-seq from both cell lines. This information has been added as supplementary Fig S4. Notably, in TP53 the arginine (CGC) exchanging the proline (CCC) at codon 72 of exon 4 is a known SNP that occurs at a high frequency (>50%) in some populations (Pietsch et al., PMID: 16550160).

10. The text has to be GLOBALLY revised – mostly, shortened and cleared of highly repetitious *cliché* and unjustified statements. For example, “transcriptional reprogramming” is repeated 22 times. Also, I feel that “3D organization of the genome” is overused. This is just a cliché that is mostly unsubstantiated in this particular paper (although “3D” is mentioned ~10 times in the main text).

The text has been revised according to the reviewer’s suggestion.

11. The references are duplicated: (44) = (46) and (45) = (47)

44. Bao, F., LoVerso, P. R., Fisk, J. N., Zhurkin, V. B. & Cui, F. P53 Binding Sites in Normal and Cancer Cells

Are Characterized By Distinct Chromatin Context. *Cell Cycle* 16, 2073–2085 (2017).

45. Botcheva, K., McCorkle, S. R., McCombie, W. R., Dunn, J. J. & Anderson, C. W. Distinct p53 genomic

binding patterns in normal and cancer-derived human cells. *Cell Cycle* 10, 4237–4249 (2011).

46. Bao, F., LoVerso, P. R., Fisk, J. N., Zhurkin, V. B. & Cui, F. p53 binding sites in normal and cancer cells

are characterized by distinct chromatin context. *Cell Cycle* 16, 2073–2085 (2017).

47. Botcheva, K., McCorkle, S. R., McCombie, W. R., Dunn, J. J. & Anderson, C. W. Distinct p53 genomic

binding patterns in normal and cancer-derived human cells. *Cell Cycle* 10, 4237–4249 (2011).

We thank the reviewer for identifying these mistakes; now corrected.

Reviewer #2:

Transcriptional dysregulation is a near-universal feature of carcinogenesis, but the molecular details underlying cancer-promoting transcriptional reprogramming are, in most cases, poorly understood. Here, the authors describe large-scale changes in chromatin accessibility and gene expression in cells expressing oncogenic HRas-G12V, focusing on redistribution of the transcriptional regulators CTCF and P53 as a potential

driving force for the chromatin and transcription alterations observed. The authors show that differential chromatin accessibility occurs within stable TADs at tested genes using 4C, and that differentially accessible regions harbor motifs for a number of interesting factors, including CTCF and P53 (as well as oncogenic ETS factors). The results are intriguing, suggesting roles for two well-known transcriptional regulators in the transcriptional rewiring that accompanies G12V transformation, and should stimulate further mechanistic investigation of carcinogenic transcriptional reprogramming. However, the presentation of the data is rather haphazard, with confusing terminology as I elaborate on below. Also, the analysis of the ChIP-seq data is rather narrow, focusing only on regions specific to one cell type or the other – I think a broader analysis of the data is needed.

Specific comments:

Reference 10 (The ENCODE Nature paper) is provided as a reference for the capacity of histone modifications to demarcate active enhancers. However, several publications prior to this one showed that H3K4me1/H3K27ac is a signature of active enhancers: Creighton et al, PNAS 2010 (PMID 21106759); Rada-Iglesias et al, Nature 2011 (PMID 21160473); Zentner et al, Genome Res 2011 (PMID 21632746).

We thank reviewer for pointing this out. We have included the mentioned publications in the revised manuscript.

Statistical analysis should be included for the phenotyping data presented in Fig. 1 and 6.

This was corrected. Thank you.

The authors interchangeably use gained and lost DARs with G12V-unique and MCF10A-unique. I found this confusing throughout, with one example being Fig. 5C, where the label uses the G12V/MCF10A nomenclature while the legend refers to gained

and lost DARs. Adding to the confusion is the use of “parental” to refer to non-transformed MCF10A. This terminology should be made consistent throughout.

We regret lack of clarity. For uniformity, the cell types are referred to as MCF10A and G12V. The differential accessible loci are referred to as DARs.

Fig. 5C refers to “CTCT” binding sites, which I assume should be “CTCF.”

We thank the reviewer for identifying the typo; now corrected.

I’d like to see some expansion of the analysis of gained and lost CTCF and P53 binding sites. I understand that the authors performed differential peak calling using MACS, but it would be interesting to **see heatmaps** generated using composite lists of peaks regardless of whether they are common or unique to MCF10A or G12V cells. This would give an idea of how complete the gain or loss of signal at these regions is. As it stands, for CTCF, the single example shown in track form in Fig. 5D suggests that there is some CTCF binding at this region in MCF10A cells. For P53, it appears that it may be a complete loss of binding. I think that heatmapping the ChIP-seq and ATAC-seq data in this manner would give the reader a more complete view of the data.

Following the reviewer suggestion, we have included heatmaps for p53 and CTCF ChIP-seq in figures 5 and 6.

In Fig. 6E, it is shown that lost DARs are enriched in P53 binding; that is, MCF10A-specific accessible sites are P53-bound.

The analysis was performed by calculating the co-localization of the indicated DARs with MCF10A and G12V combined ChIP-seq dataset. This is now indicated in the figure legend.

In the corresponding text, it is stated that this result confirms the direct role of P53 in reprogramming transcription in G12V cells. It seems more **that this suggests** a suppression of transcriptional rewiring, as these sites are lost in G12V cells (at least in the 1 example shown in Fig. 6F).

This valid point was addressed and clarified in the manuscript. The revised corresponding text states that "... , reinforcing that rewiring of P53 regulatory network contribute to transcriptional reprogramming in in G12V cells ".

Again, it would be helpful to see how widespread the loss of P53 binding really is at lost DARs.

The link between the loss of p53 binding and lost DARs is now demonstrated in the heatmap of p53 ChIP-seq (bottom group, Fig. 6G). Quantitatively, the proportion of chromatin accessibility was similar in MCF10A and MCF10A-specific p53 binding loci (22%, 441/2007; 20% 112/565, respectively), and in line with recent measurements in GM06170 fibroblasts (PMID 28973438; 21% 565/2,638). Yet, the loss of chromatin accessibility following HRas G12V-induced transformation was remarkably high in MCF10A-specific p53 binding loci (85%, 95 lost DARs/112 constitutively accessible), relative to the global loss of accessibility in p53 binding loci (56%, 247 lost DARs /441 constitutively accessible, $p=3.872e-08$ proportional test), which by itself was greater than the global loss of chromatin accessibility (18%, 7,589 lost DARs /42,546 constitutively accessible, $p< 2.2e-16$ proportional test). Thus the loss of chromatin accessibility in MCF10A p53 binding loci is specific. This information together with the figure has been included in the manuscript (Supplementary Fig. S6)

Have the authors deposited their data in GEO? If so, an accession number and reviewer token should be provided for confirmation. The authors should make any custom scripts used for analysis available on Github or another repository.

We regret the inconvenience. The GEO number was indicated in the submission form but accidentally omitted from the manuscript.

The GEO accession is GSE140254:

<https://eur02.safelinks.protection.outlook.com/?url=https%3A%2F%2Fwww.ncbi.nlm.nih.gov%2Fgeo%2Fquery%2Facc.cgi%3Facc%3DGSE140254&data=02%7C01%7Cavital.sarusi%40biu.ac.il%7C567b68b1e3fe462343c208d76859e8c2%7C61234e145b874b67ac198feaa8ba8f12%7C0%7C0%7C637092607703499784&sdata=JKNS7zdU50GVn5WutcRAQXOAG4I2z5Srs4YAtKKHtEk%3D&reserved=0>

Token: qlgjaskwrjsmap

No custom scripts have been used.

Reviewer #3

The authors over-expressed a mutated HRas (G12V) in an epithelial cell line MCF10A and explored the potential mechanism that drives the transcriptional reprogramming.

They compared gene expression (RNA-Seq) and chromatin accessibility (ATAC-Seq) between the two lines (MCF10A vs G12V). The expression changes match phenotype changes induced by HRas, relating upregulated genes to invasion and downregulated genes to apoptosis (Fig 1 & Fig 2). The ATAC-Seq data analysis revealed changes in accessibility at tens of thousands of regulatory sites (differentially accessible regions; DARs), correlated well with expression changes (Fig 3). To predict target genes for distal regulatory sites, the authors first showed that the up-regulated genes and the down-regulated genes do not likely reside in the same topologically associating domain (TAD). They further showed that cell-type specific DARs are enriched within TADs containing cell-type specific expressed genes (Fig 4). They author then narrowed down to DARs that are associated with differential expressed genes within the same TADs, carried out motif enrichment analysis, identified potential regulators including CTCF (Fig 5) and P53 (Fig 6), and confirmed the bindings by ChIP-Seq analysis. The two transcription factors exhibited a redistribution of binding as induced by HRas without a change in protein level. The authors also confirmed the transcription response induced by P53 (Fig 6). Many of the observations are predictable based on literature (especially for Figs 2,3 and 4). The findings on Figure 5 and 6 require further evidence to understand the mechanism regarding the redistribution of TF binding induced by HRas.

1) The authors used TAD to connect DAR to differentially expressed (DE) genes. Details of the method are missing. For example, how would the author assign the target(s) of a DAR, if a TAD contains multiple DE genes and multiple DARs?

Indeed, TADs may contain several genes and we found that the direction of the transformation-induced transcriptional variation within TADs is overall uniform (Fig. 4C). In addition, genes may be regulated by several regulatory loci located in the same TAD. Therefore, TADs were used as genomic units in which genes and regulatory elements associate together and the differential accessible sites within up or down TADs were used collectively for the analyses. We have clarified this point in the methods section:

“All the DARs within up or down TADs were used collectively for downstream analyses”.

2) The benefit of using TAD to choose subsets of DARs for motif analysis **is unclear**. Would applying motif analysis to non-promoter DARs without considering TAD yield similar hits to those shown in Fig 5A and Fig 6A. In addition, as separating DARs based on upregulated genes and downregulated genes gave similar motifs with comparable % of target sequences (Fig 5A and Fig 6A), the benefit of considering expression changes seems marginal.

We thank the reviewer for this comment. Motif discovery analysis for the entire DAR dataset uncovered the binding motifs of CTCF and p53 in 8.97% and 10.90% of the gained and lost DARs, respectively. This information has been added to the manuscript.

3) The author showed that cell type specific DARs are enriched within TADs containing cell-type specific expressed genes. However, the data presentation for Figure 4D is confusing. It is difficult to figure out the steps of the data analysis that has led to Figure 4D, even though the conclusion fits expectation.

*The figure shows the density of ATAC-seq peaks in MCF10A and G12V cells within TADs of differentially expressed (DE) genes. The peak densities were calculated for the same two sets of genomic regions defined by TADs of up- or down-regulated genes. The text in the figure was modified to clarify this point. The figure legend now indicates that:” (D) Boxplots showing ATAC-seq peak density in domains of up- (red) or down regulated genes (blue). Left – MCF10A regulatory sites, right- G12V regulatory sites. * $p < 0.01$, Wilcox test. “.*

4) The data analysis for regulator inference in Fig 2E is not clear. Did the author use differentially expressed genes as an input or use a pre-ranked list of all genes? I am surprised to see that LPS (lipid) and dexamethssone (a drug) were predicted as two regulators.

The analysis was performed by Ingenuity Pathway Analysis software (IPA) which finds enrichment of pathways based on the input genes against a knowledge base. As input we used the lists of differentially expressed (DE) genes with their logFC values, that add statistical power the IPA analysis and provide the directionality which guides the genes that dictate the pathway. IPA prediction of the factors that may govern the pathways is presented in figure 2E. No additional analysis was done. The information from IPA is valuable when supported by and supports further experimental data, as in the case of p53.

The other mentioned pathways represent increase in positive regulators of the immune system such as TNF, IFNG and LPS (lipopolysaccharides are components of the outer membrane of gram-negative bacteria that activate the immune system), together with decrease in immunosuppressive pathways such as Dexamethasone. Despite the relevance of the immune gene signature in cancer, transcriptional regulators of immune genes did not emerge in our genomic analysis.

5) Figure 5C: the author need to include DARs shared by G12V and MCF10A as a control.

The overlap of ATAC-seq peaks shared by G12V and MCF10A with CTCF was added to figure 5D in the revised manuscript.

6) The authors stated a global redistribution of CTCF binding but only included a specific example (Figure 5D) **without detailed global analysis**. They showed that the number of CTCF peaks increased from 22K to 31K from MCF10A to C12V. But the number of peaks would depend on sequence depth. A differential peak analysis similar to Figs 3A and 3C will help to clarify (also apply to the P53 ChIP-Seq data analysis).

As suggested by the reviewer, we have added a detailed analysis of the ChIP-seq experiments. Heatmaps of ChIP-seq with ATAC-seq were added to figures 5, 6 and MA plots were added as supplementary figure S5.

7) Fig 5D and 6F: are the two regions functional remarkable. Globally, what are the functions for all the differential CTCF binding and P53 binding sites. A GREAT analysis (<http://great.stanford.edu/public/html/>) would give some hints.

The point raised by the reviewer, uncovering functions which may be regulated by the variable transcription factor binding loci, is an excellent one. GREAT links genes to regulatory elements co-residing in spatial domains. The analysis we performed by IPA (formerly figure 6G, currently supplementary figure S6) was based on similar principles but took into account also the variance in gene expression. We analyzed cell type-specific genes which are linked in spatial domains to cell type -specific transcription factor binding loci. For uniform and relevant presentation, the figure includes only disease functions.

As suggested by the reviewer we performed GREAT analysis for the DE peaks determined by EdgeR. This analysis revealed that MCF10A-specific p53 and CTCF peaks retrieved “Human Phenotype” and “Biological Process” terms, respectively. Both terms are related to development and differentiation functions. Similar results were retrieved by IPA. These terms are in line with the loss of cellular identity and increased plasticity in cancer progression.

GREAT analysis of G12V-specific p53 peaks retrieved multiple Biological Process, cancer related terms including signal transduction by p53, G1-S transition, mitotic checkpoint and DNA damage response. These terms were among more than hundred “canonical” terms retrieved from IPA. The G12V-specific CTCF peaks did not retrieve any GO term.

We have now added the GREAT analysis for p53 ChIP, which provides biological processes terms, together with disease related terms from IPA analysis to Supplementary table S3.

8) The authors observed a redistribution of CTCF or P53 binding but did not investigate **the mechanisms**. From motif analysis, the authors hypothesized that TFs recognizing

AP1 motif could involve in the redistribution of P53 binding. I would suggest the author to verify the hypothesis by doing a **CRISPR KO at an AP1 site and measured the binding change of P53 on the site in G12V by CHIP-PCR.**

Based on the enrichment for AP1 binding motif in G12V-specific p53 loci and its pioneering capacity, AP1 binding was suggested as a possible mechanism for p53 binding in G12V cells. The experiment suggested by the reviewer should be performed in a larger scale, that is beyond the scope of the current study.

Interestingly, ATF3 BATF3 and FOSL1 members of the AP1 family of transcription factors are induced in G12V cells. ATF3 is known to repress pro-apoptotic genes in p53 pathway (PMID: 22046379). FOSL1 is highly expressed in mutant KRAS lung and pancreatic cancers and it is a positive regulator of proliferation and viability of triple negative breast cancer (PMID: 28616588, PMID: 29273624).

9) Some minors: some of the labeling of the subpanels in Fig 6 are not correct; I assumed Fig 6D to be Figure 6B. Figure 5D: scale on the y-axis is missing. GSE# is missing.

This was corrected. Thank you.

Reviewers' Comments:

Reviewer #1:

Remarks to the Author:

The paper has been significantly improved, and can be accepted for publication.

Reviewer #2:

Remarks to the Author:

The authors have pretty well addressed my initial concerns, and I have no major points left that I'd like to be addressed. I just have one suggestion in terms of data display: for clustered heatmaps, it would be useful to list the number of sites in each cluster, for a handy reference to how many sites display differential CTCF or P53 binding.

Reviewer #3:

Remarks to the Author:

The authors have satisfactorily addressed my concerns. Two minor left:

1) Fig2E: according to the response, the analysis was to check the enrichment of DE genes in predefined gene sets associated with certain features, for example target genes of TP53, genes responsive to Dex, genes responsive to LPS, etc. Calling the features all as regulators is misleading: for example, LPS does not target promoters or enhancers, and is not a transcription regulator.

2) The authors chose the functional validation on the recruitment of P53 through AP-1 binding TFs for future investigation. Interestingly, they observed expression up-regulation of several AP-1 binding TFs including ATF3, FOSL1, and BATF. My search in the protein-protein interaction DB STRING indicates physic interaction between ATF3 and P53. The authors may include these observations in discussion.

Reviewer #2 :

1. I just have one suggestion in terms of data display: for clustered heatmaps, it would be useful to list the number of sites in each cluster, for a handy reference to how many sites display differential CTCF or P53 binding.

The number of sites in each cluster is now included in figures 5 and 6.

Reviewer #3:

1) Fig2E: according to the response, the analysis was to check the enrichment of DE genes in predefined gene sets associated with certain features, for example target genes of TP53, genes responsive to Dex, genes responsive to LPS, etc. Calling the features all as regulators is misleading: for example, LPS does not target promoters or enhancers, and is not a transcription regulator.

The wording was modified to factors that can explain gene expression changes.

2) The authors chose the functional validation on the recruitment of P53 through AP-1 binding TFs for future investigation. Interestingly, they observed expression up-regulation of several AP-1 binding TFs including ATF3, FOSL1, and BATF. My search in the protein-protein interaction DB STRING indicates physic interaction between ATF3 and P53. The authors may include these observations in discussion.

We thank the reviewer for this suggestion. The following text was added to the discussion" In addition, ATF3 and FOSL1 subunits of the AP-1 transcription factor complex that were up-regulated in G12V cells, were shown to have regulatory interactions with p53^{55,56"} .